# Comparing the Efficacy of a Microperforated Titanium Membrane for Guided Bone Regeneration with an Existing Mesh Retainer in Dog Mandibles

**DOI:** 10.3390/ma14123358

**Published:** 2021-06-17

**Authors:** Hiroshi Hasegawa, Tetsuharu Kaneko, Manabu Endo, Chihiro Kanno, Morio Yamazaki, Sadanoshin Yaginuma, Hiroki Igarashi, Hideaki Honma, Seiichiro Masui, Mizuki Suto, Yukihiko Sakisaka, Hiroshi Ishihata

**Affiliations:** 1Department of Oral Surgery and Dentistry, Fukushima Medical University, 1, Hikariga-oka, Fukushima 960-1295, Japan; tetsu1277@yahoo.co.jp (T.K.); manabuendo0926@gmail.com (M.E.); chihiri@fmu.ac.jp (C.K.); 0624m.yamazaki@gmail.com (M.Y.); sadashin@fmu.ac.jp (S.Y.); 2Department of Dentistry and Oral Surgery, Koseikai Hospital, 16-2, Nariide, Kitasawamata, Fukushima 960-8251, Japan; nayacisan@gmail.com; 3Department of Dentistry and Oral Surgery, Omachi Hospital, 3-97, Omachi, Haranomachi-ku, Minami-Soma, Fukushima 975-0001, Japan; h.honma.mail@gmail.com; 4Clinical Research Center, Fukushima Medical University Hospital, 1, Hikariga-oka, Fukushima 960-1295, Japan; semasui@fmu.ac.jp; 5Division of Periodontology and Endodontology, Department of Ecological Dentistry, Tohoku University Graduate School of Dentistry, Sendai 980-8575, Japan; mizuki-s@dent.tohoku.ac.jp (M.S.); yuki@dent.tohoku.ac.jp (Y.S.); hiroshi.ishihata.a8@tohoku.ac.jp (H.I.)

**Keywords:** guided bone regeneration, titanium membrane, barrier membrane, microporous titanium

## Abstract

Acute-type lateral ridge defects (25 mm × 6 mm × 5 mm) were bilaterally created in the mandibles of four dogs (two defects per animal). The defects were reconstructed with particulate autologous bone and covered with a microperforated titanium membrane (Ti-honeycomb membrane, TiHM) or an existing conventional titanium mesh as control. The samples were dissected after 16 weeks postoperatively and processed for radiographic, histologic, and histomorphometric analyses. Regenerated tissue and bone volume were significantly larger in the TiHM group than in the control group (*p* = 0.05; *p* = 0.049). In contrast, bone mineral density was similar between the two groups. Histomorphometric analysis revealed that the regenerated bone area and calcific osseous area were larger in the TiHM group than in the control group; however, the differences were not significant. The efficacy of TiHM was generally satisfactory with the potential to become a standard tool for the GBR procedure; however, early membrane exposure will be a major problem to overcome.

## 1. Introduction

In guided bone regeneration (GBR), a membrane covering the bone defect and filling materials plays an important role, thus significantly affecting the treatment outcome. According to the type of biomaterial, membranes are classified as resorbable or non-resorbable [1]. The chief requirements for the membrane are described as the following: (1) biocompatibility; (2) mechanical properties for retaining the filling and spacemaking to create a space for osteogenic regeneration; (3) clinical manageability; (4) cell occlusivity; (5) tissue integration for stability with respect to the surrounding tissue [2,3].

Titanium has a high biocompatibility as well as a high weight tolerance of physical loads for spacemaking. Therefore, titanium membranes processed into thin sheets have the possibility to meet the aforementioned requirements when manufactured into various shapes or sizes. Titanium mesh, a non-resorbable membrane, has been largely used in oral, maxillofacial, plastic and reconstructive surgery, as well as GBR procedures. The available titanium meshes are more than 100 μm thick, and have multiple pores of 1–3-mm diameter [4,5,6,7]. The thickness provides sufficient stiffness to the mesh, thus enabling it to protect the inner space from the external forces of mastication by rigid fixation with screws. Therefore, the mesh can be applied to a large bone defect after mandibulectomy because of its stiffness [8].

FRIOS^®^ Boneshield (FRIOS^®^ Boneshield; Dentsply Sirona Corp., Tokyo, Japan), which is a thin film-like titanium membrane, has been used globally as a more flexible retainer for enveloping the filling. With a thickness of approximately 20 μm and micropores (diameter, 40–60 μm; pitch ≥250 μm) that allow the free diffusion of nutrients and tissue fluid, FBS is easy to handle because of its shape and is significantly effective in the clinical setting [9,10,11,12,13]. However, because it is not possible to avoid fibrous tissue ingrowth from outside the membrane, the barrier function at such pore sizes would be suboptimal [14].

Recently, a new pure titanium membrane, Ti-honeycomb membrane (TiHM, Morita Co., Osaka, Japan) was approved as the first improved product of FBS in Japan. Laser processing technology was used to create a 20-μm thick pure titanium membrane with denser and smaller microperforations (diameter, 20 μm; pitch, 50 μm) [15]. This ultrafine structure was expected not only to minimize the ingrowth of soft tissue into the bone defect while maintaining nutrients and tissue fluid permeability across the membrane, but also to promote the attachment, migration, and proliferation of bone marrow-derived cells as a scaffold in an in vitro study [16]. In addition, the structure may facilitate bone regeneration through the same mechanism as in vivo studies using autologous bone and beta-tricalcium phosphate. Furthermore, in vivo studies demonstrated that TiHM was not inferior to FBS in terms of safety and efficacy [17,18].

However, no study has compared the efficacy of TiHM with that of a conventional titanium mesh. Therefore, it has become necessary to investigate the efficacy of these two membranes with different thicknesses and pore sizes. In the present study, we used the mandibles of beagle dogs to assess the efficacies of TiHM and titanium mesh in alveolar bone defects using autogenous bone.

## 2. Materials and Methods

### 2.1. Fabrication of Microperforated Titanium Membrane

A microperforated TiHM structure (size L, 27 mm × 42 mm with a frame) was designed to be formed into a precise and unified piercing array on a thin titanium sheet. The ablation piercing process used a short pulsed laser with a nanosecond pulse duration, to produce a hexagonal pored section that includes a uniform, staggered array of ordered mircon-sized holes (Figure 1). The irradiation power and focus of the laser beam were optimized to penetrate through the thin pure titanium sheet as quickly as possible, to accelerate serial perforation and prevent thermal degeneration of the material (Lastech Co., Ltd., Fujimi, Japan). A heat-affected zone along the cut edge of the round hole, which was produced after the laser process, remained as a thin layer of dross. Thus, after completing the piercing process, the workpiece was etched by immersion into a hydrofluoric acid solution.

A conventional laser processing system that used an oscillating, nanosecond pulse frequency light source was optically tuned to confirm an output with a focused diameter that was within 20 microns. Galvano-drive mirrors reflected and emitted the beam to the workpiece and oriented the beam to scan the surface area over a range of a few square millimeters. The penetration processing velocity with a 20 µm precise diameter laser was 80 holes per second. When the Galvano-drive mirror serial piercing process completed over the scannable range, a motorized X-Y stage moved the workpiece horizontally by a few millimeters. Then, the laser piercing process was performed under programmable control in the neighboring area.

The sample material was manufactured with dense microperforations (diameter, 20 μm; pitch, 50 μm) within regular hexagonal non-perforated compartments (inscribed diameter, 1.0 mm; width, 0.2 mm), on a pure titanium 20 μm-thick sheet with a honeycomb shape (Figure 2a,b). The titanium reinforcement frame that imparted maintenance properties for the spacemaking was joined on the membrane by multiple spot welding (Hayashi Seiki Seizou Co., Ltd., Sukagawa, Japan).

An ultra-flex mesh plate (UFMP, 35 mm × 48 mm, 100-μm thickness, Kyocera Co., Kyoto, Japan) was used as the standard thinner titanium mesh in this study and had star-like pores with a circumscribed diameter of approximately 3200 μm, inscribed diameter of approximately 1600 μm. These pores enabled rigid screw fixation (Figure 2c).

### 2.2. Animals

This study included four male beagle dogs aged 10–14 months (weight, 10–14 kg). After a 1-week acclimatization period, each animal was randomly assigned a number. The experiment was conducted at a facility certified by the Association for Assessment and Accreditation of Laboratory Animal Care International, and the protocol was approved by the Animal Care and Use Committee of the same facility (approval number, IACUC714-032).

### 2.3. Study Design

Standardized box-shaped defects were created in the buccal aspects of the bilateral mandibular bone of each dog (two defects per animal). Grafting was performed using particulate autogenous bone (PAB) harvested from the iliac bone. The graft on one side was covered with TiHM (TiHM group [*n* = 4]) and on the other side, in the same animal, with UFMP (UFMP group [*n* = 4]). Membrane removal was performed before autopsy at 16 weeks after surgery to evaluate the membrane and surrounding tissue’s postoperative conditions. The harvested tissue was radiographically and histologically examined in order to assess each membrane’s efficacy.

### 2.4. Surgical Procedure

Intramuscular medetomidine hydrochloride 0.08 mg/kg (Domitor, Orion Corporation, Espoo, Finland) and midazolam 0.4 mg/kg (Dormicum, Astellas Pharma Inc., Tokyo, Japan) were administered in order to induce anesthesia. Following sedation, the dogs were endotracheally intubated, and anesthesia was maintained using 0.5–5.0% isoflurane (Forane, Abbott Japan Inc., Tokyo, Japan).

Local anesthesia using 2% lidocaine (Xylocaine, Dentsply Sirona Corp., Tokyo, Japan) was administered from the first mandibular premolar (P1) to the second mandibular molar (M2). Mucosal incisions were performed, one along the P1 to M2 cervical region, followed by vertical incisions at P1 and M2. The mucoperiosteal flap was elevated, and the second (P2), third (P3), and fourth premolar (P4), as well as the first molar (M1) were extracted. Subsequently, using a dental fissure burr (Osada portable unit “Daisy”, Osada Electric Industry Co., Ltd., Tokyo, Japan), a box-shaped defect (25-mm mesiodistal width; 6-mm depth; 5-mm depth from the buccoalveolar bone surface) was created while continuing irrigation with physiological saline (Figure 3a).

A dental caliper was used to measure the distances from the distal end of the P1 tooth cervix to the mesial side of the bone defect as well as the center of the mental foramen. It was also used to measure the distance from the bone defect’s distal side to the mesial end of the M2 tooth cervix. These distances were used as references for micro-computed tomography (micro-CT) analysis.

About 3 g of PAB, including the cortical bone from the left wing of the ilium, was simultaneously harvested during the GBR procedure, following which it was pulverized with a bone mill (YDM Corp., Tokyo, Japan).

After the PAB was densely packed into the bone defect (Figure 3b), the trimmed TiHM or UFMP was bent at angles of 90° to completely cover the defect and PAB. Titanium screws (diameter: 1.4 mm, length: 3 mm; Le Forte System. Jeil Medical Corp., Seoul, Korea) were used to secure the membrane at six sites on the buccal side and two sites on the alveolar crest of the bone (Figure 3c,d). Suturing was performed using 5-0 absorbable sutures. All surgeries were performed by a single surgeon, and the membrane position was confirmed immediately following each procedure using dental radiographs.

Postoperatively, dihydrostreptomycin sulfate-benzylpenicillin procaine (Mycillin Sol Meiji, Meiji Seika Pharma Co., Ltd., Tokyo, Japan) was administered intramuscularly, once a day for 3 days, at a volume of 0.05 mL/kg. In addition, ketoprofen (Capisten, Kissei Pharmaceutical Co., Ltd., Tokyo, Japan) was administered at a dose of 2 mg/kg once a day for two days. For 2 weeks following surgery, the animals were fed with a mixture of equal amounts solid feed and drinking water, daily. Starting from the third postoperative week, solid feed was provided.

### 2.5. Clinical Observation

The wounds were observed at 1, 2, 3, 4, 6, 8, 10, and 12 weeks postoperatively to identify postoperative complications such as inflammatory symptoms and membrane exposure.

### 2.6. Membrane Removal Surgery

Sixteen weeks postoperatively, before being sacrificed, all animals underwent membrane removal. The titanium membranes were removed under general anesthesia, using the same approach as that used for the GBR procedure. During the procedure, the membrane conditions, adhesion to the surrounding tissue, and regenerated bone were observed.

### 2.7. Retrieval of Specimens

After being administered an intravenous dose of 0.5 mL/kg pentobarbital (Pentobarbital Sodium Salt, Tokyo Chemical Industry Co., Ltd., Tokyo, Japan), the animals were all sacrificed via exsanguination. Then, for perfusion fixation, 10% neutral-buffered formalin solution 30 mL was injected into the bilateral carotid arteries. This was followed by removal and resection of the bilateral mandibles along the medial line. The specimens were next fixed in 10% neutral-buffered formalin solution for a total of 7 days and, to evaluate bone regeneration, were subjected to radiographic, histological, and histomorphometric analyses.

### 2.8. Radiographic Analysis

The animal specimens were imaged using micro-CT (SKYSCAN1174-KF; Bruker Corp., Billerica, MA, USA) at a voltage of 50 kVp, a current of 800 μA, a slice thickness of 19.7 μm/pixel, and a matrix size of 1304 × 1024. Using a bone mineral density (BMD) phantom (Medium BMD phantom SP-4003; Bruker Corp.), the specimens were also imaged under the same conditions to set a reference value for BMD. NRecon software (ver. 1.6.9.18; Bruker Corp.) was used to reconstruct the images in order to obtain volumetric imaging data, and CTVox (ver. 3.0; Bruker Corp.) was used to create three-dimensional (3D) images. The inclination of the 3D images was adjusted, and the analytical range was set using DataViewer (ver. 1.5.1.2; Bruker Corp.). 

CTAn (ver. 1.15; Bruker Corp.) was used to perform 3D analysis, and to distinguish mineralized tissue, the Otsu algorithm was used to select a threshold of 659 mgHA/mL across all samples. First, the mid-section of the bone defect was determined on the 3D images by referring to the distance from the cervical portion of P1 and M2, and the center of the mental foramen to the edge of the bone defect measured during surgery, as well as the dental radiographs taken immediately after surgery. The two sections were subsequently identified as the mesio-distal limit of the bone defect, 12 mm away from the mid-section, mesially and distally. After identifying the border between the bone defect and the host bone in these sections, the range surrounded by the linear margin of the bone defect (6 mm in height with a 5-mm base) and perimeter of the newly formed bone was manually outlined as the region of interest (ROI). Furthermore, the volume interpolated between these ROIs was referred to as the volume of interest (Figure 4). Following that, tissue volume (TV), bone volume (BV), bone volume-to-tissue volume ratio (BV/TV ratio), and BMD were calculated within the bone defect as radiological morphometric indices [19]. Micro-CT analysis was performed as previously described [17,18].

### 2.9. Specimen Preparation

The mandible was split bilaterally using a handsaw buccolingually at the center of the surgical site. The specimens were dehydrated in increasing concentrations of alcohol solution and embedded in polymethyl methacrylate (MMA; Wako Pure Chemical Industries Ltd., Osaka, Japan). Polished, undecalcified, 20–50 μm-thick specimens were prepared from the mesial blocks. The sections were stained with toluidine blue (TB), which stains mineralized bone tissue for histomorphometric analysis.

### 2.10. Histomorphometric Analysis

A digital camera attached to a microscope (DS-Fil-L2; Nikon, Tokyo, Japan) was used to collect images of all TB-stained sections. The regenerated bone area (RBA) and calcific osseous area (COA) were measured using Area Q (S-tech Ltd., Tokyo, Japan). RBA was measured by tracing the perimeter of the newly formed bone. The border between the original host bone and the regenerated cortical bone was determined by the difference in shading in the TB staining (Figure 5) [17,20]. COA was determined by outlining each segment of the TB-stained bone tissue, followed by summation of the cortical bone and cancellous bone within the RBA. The calcific osseous rate (COR) was calculated as the percentage of COA to RBA.

### 2.11. Statistical Analysis

The radiological morphometry indices on micro-CT and histomorphometric indices were compared between the two groups (*n* = 4 each) using an unpaired *t*-test. Values of *p* < 0.05 were considered statistically significant on both sides. SPSS Statistics 23 (IBM Japan, Tokyo, Japan) was used for statistical analysis.

## 3. Results

### 3.1. Clinical Evaluation

During the healing process, membrane exposure was found at four sites as follows. In the TiHM group, membrane exposure was found at one site, three weeks postoperatively, and the membrane was removed at nine weeks postoperatively because the exposure size was enlarged (Figure 6a–c). At another site, although membrane exposure was found around a secured screw eight weeks postoperatively, removal was not required because the exposure did not enlarge.

In the UFMP group, secured screws and the surrounding membrane exposure were found at two sites on the alveolar crest of two animals at eight weeks postoperatively. However, removal was not required because the exposures did not enlarge (Figure 6d).

During the removal surgery in the TiHM group, no membrane adhesion to the mucoperiosteal flap or the underside tissue was observed. In addition, the membrane was able to be peeled back with ease at all sites. The tissue under the membrane had been replaced with bone tissue, and the surfaces that were in contact with the laser-perforated sections indicated as dark red in color with a honeycomb shape. (Figure 7a,b).

On the other hand, in the UFMP group because of the adhesion of the membrane to the surrounding tissue (Figure 8a), it was difficult to detach the tissue from the membrane at all sites (Figure 8b,c). Thus, the UFMP had to be forcefully removed leaving a scar-like tissue; however, regenerated bone was found beneath the soft tissue similar to that in the TiHM group (Figure 8d).

### 3.2. Radiographic Evaluation

The mean TV, BV, BV/TV ratio, and BMD in the TiHM group were 290.4 ± 14.3 mm^3^, 192.1 ± 16.3 mm^3^, 66.45 ± 8.4%, and 0.85 ± 0.04 g/cm^3^, whereas in the UFMP group they were 241.9 ± 37.0 mm^3^, 144.4 ± 35.0 mm^3^, 59.2 ± 6.4%, and 0.85 ± 0.03 g/cm^3^, respectively. Thus, the TV, BV, and BV/TV ratio were larger in the TiHM group than in the UFMP group (Figure 9a,b). Furthermore, the former two indices showed significant differences (*p* = 0.05; *p* = 0.049). However, the BMD value was similar in both the groups (Figure 9c).

### 3.3. Histomorphometric Analysis

As shown in Figure 10, the respective mean RBA, COA, and COR were 21.8 ± 3.4 mm^2^, 12.5 ± 1.9 mm^2^, and 57.4 ± 5.8% in the TiHM group and 19.4 ± 2.3 mm^2^, 9.3 ± 1.8 mm^2^, and 47.9 ± 7.9% in the UFMP group, respectively. The values were larger in the TiHM group than in the UFMP group, although no statistical differences were found.

## 4. Discussion

TiHM of the membrane and UFMP of the mesh essentially differ in their thickness and pore size, which are 20 μm and 20 μm, respectively, in the former, and more than 100 μm and 1000–3000 μm in the latter, respectively. Although the specification of these two types of titanium products is different on a dimension of porous structure, they could be both applied for retaining the autograft bone at the donor site on GBR operation. The porous structure of these retainers could maintain a permeability for the incoming nutrient-containing fluid from surrounding tissues. The other type of retainer for the surgical operation of GBR is a resorbable membrane fabricated from a biodegradable polymer material [1]. The characteristics of the resorbable retainer are relatively soft and flexible and have a permeability when the material is a fabric knitted with fibers. The newly developed titanium membrane of TiHM was designed to eliminate the high rigidity that is the physical property of existing titanium mesh, and to obtain the plasticity and flexibility for the utility of formation of fitting to the operation site. To provide sufficient tissue fluid permeability to the thin titanium sheet, laser processing technology for precise piercing in micron order was possible to effectively utilize the preparation of numerous through holes in a short period of working time.

UFMP of the titanium mesh is superior to TiHM in terms of strength and stiffness, and the screw fixation through the pore is steadier and more stable. On the other hand, TiHM is thinner, which is easy to handle and enables the formation of a fine form that contours the bone defect. Moreover, the edge of the membrane after trimming is less likely to injure the oral mucosa. However, screw fixation is less steady because the fixation is to a thin membrane. Thus, it is evident that these two membranes have different efficacies because of their variable features, although they are made from the same material, titanium.

When using the titanium membrane, including the TiHM, FBS, and meshes, infection by membrane exposure is a major concern. In the literature, the membrane exposure rate and success rate of bone augmentation were 0–33% and 91–100% when using FBS, and 0–51% and 75–100%, respectively, when using meshes [4,5,6,7,9,10,11,12,13,21,22,23]. Moreover, the occurrence of membrane exposure was also dependent on the procedure performed; it was especially higher for vertical bone augmentation [9,21]. However, it was reported that most exposed titanium membranes or meshes did not necessarily require removal because of less bacterial adhesion and acceptable cleaning of material; meanwhile, bone augmentation was likely to fail in early membrane exposure such as less than a few weeks postoperatively [5,6,9,10,21,22,23,24].

In this study, early membrane exposure at three weeks postoperatively was found at one site in the TiHM group, in which the membrane was removed because of the enlargement of exposure. This exposure was probably due to technical failures of suturing and may have enlarged because the animal was unable to rest adequately and perform frequent mouth care and cleaning. Other membrane exposures, including two sites in the UFMP group and one site in the TiHM group were found at eight weeks postoperatively, where the exposure was not enlarged. It was speculated that the mucosa contacting a screw became thinner and was ruptured by the contraction of the wound or external pressure of mastication; however, the exposure after stabilization of the wound would not enlarge.

In terms of efficacy, on radiographic analysis, TV and BV were significantly higher in the TiHM group than in the UFMP group. This may be because a thinner non-resorbable membrane would induce larger regenerated TV in the limited space for GBR. On the other hand, considering the pore size, the optimal size has not yet been defined in the literature [25,26]. However, a pore size greater than 100 μm has been reported to be necessary for penetration of the blood vessels from outside of the membrane [25,27]. Therefore, larger pores of the mesh may be advantageous for bone regeneration due to angiogenesis through the pores; however, they also allow the ingrowth of soft tissue other than the blood vessels, resulting in the disruption of barrier function. Subsequently, the larger pores could cause a decrease in the volume of regenerated bone, and an increase in the difficulty of membrane removal due to soft tissue ingrowth [4,28]. Her et al. [5] reported that the use of titanium mesh with a larger pore (≥2 mm) led to an ingrowth of a greater volume of soft tissue than that with a smaller pore size (˂2 mm). In addition, in using the former mesh, the study recommended concomitant use with a resorbable membrane to decrease soft tissue invasion. The UFMP used in the present study had a pore size of approximately 1.6–3.3 mm. However, a large amount of scar tissue around the membrane was found during membrane removal, which suggested depression of the barrier function.

On the other hand, the TiHM used in the present study had micropores of 20 μm in diameter. The length of human erythrocytes is approximately 6–8 μm, whereas that of white blood cells (excluding macrophages) is around 7–30 μm [29] Therefore, the pores could minimize the infiltration of cells from the surrounding tissue while still permitting body fluids, nutrients, and erythrocytes to pass through; however, the penetration of blood vessels was not expected because of the pore size. Furthermore, the pores are placed at a distance of 50 μm from one another, which is expected to facilitate bone regeneration as a scaffold [15,16]. In this study, after membrane removal, thin fibrous tissue was found beneath the TiHM without adhesion between the membrane and surrounding tissue, which suggested the retention of barrier function macroscopically. In a previous in vivo study [18]. the thin fibrous tissue beneath the TiHM was suspected to have originated from the host bone rather than from outside of the membrane, although further investigation on the mechanism of the fibrous tissue formation is needed. 

Thus, it is obvious that the difference in barrier function due to different pore sizes mainly affected the regenerated TV and BV. To increase the barrier function, the pore size needs to be smaller, and the membrane inevitably needs to be thinner technologically. Consequently, the membrane strength will decrease, with a decrease in membrane stability due to screw fixation to thinner membranes. Further study is needed to determine the extent to which the barrier function is required and to define the optimal pore size in a titanium membrane for a GBR procedure.

The present study suggested that the efficacy of TiHM was not inferior to that of the standard mesh. However, the experimental design had several drawbacks, such as the small number of animals, “acute-type lateral ridge defects”, and use of iliac bone with osteogenic potential. Further studies using a chronic model, or with bone filling material other than autologous bone, are needed to evaluate its efficacy compared to other standard meshes.

In clinical use, it will be important to use these membranes differently. The mesh may preferably be used for large bone augmentation due to its stiffness rather than for small bone augmentation, whereas TiHM is easy to handle and is applicable for small to moderate bone augmentation because of its less steady screw fixation, even though it has a reinforced frame. Considering the differences in these membranes, more clinical applications of TiHM are expected. Although there may be a limitation in the present study due to the use of the GBR procedure model without placing a dental implant [30,31], this membrane has the potential to become a standard tool in GBR procedures; however, the problem of early membrane exposure needs to be overcome.

## 5. Conclusions

A newly designed, laser-perforated pure titanium membrane, TiHM, was evaluated for its efficacy in comparison with a standard thin titanium mesh, in the reconstruction of mandibular bone defects with bone graft in dogs.

The present study suggested that the efficacy of the TiHM was not inferior to that of the thin mesh; however, further study is needed to evaluate the efficacy of TiHM in comparison to other meshes. Although more clinical applications need to be ascertained, TiHM has the potential to become a standard tool in GBR procedures.

## Figures and Tables

**Figure 1 materials-14-03358-f001:**
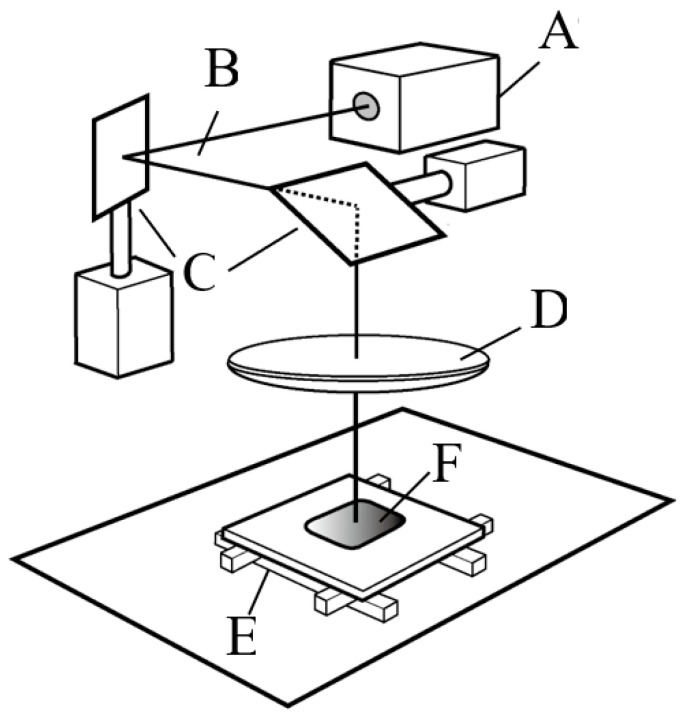
Schematics of microperforation process for titanium sheet. (**A**): Pulse laser, (**B**): Laser beam, (**C**): Galvano-mirrors, (**D**): Lens, (**E**): X-Y stage, (**F**): Workpiece.

**Figure 2 materials-14-03358-f002:**
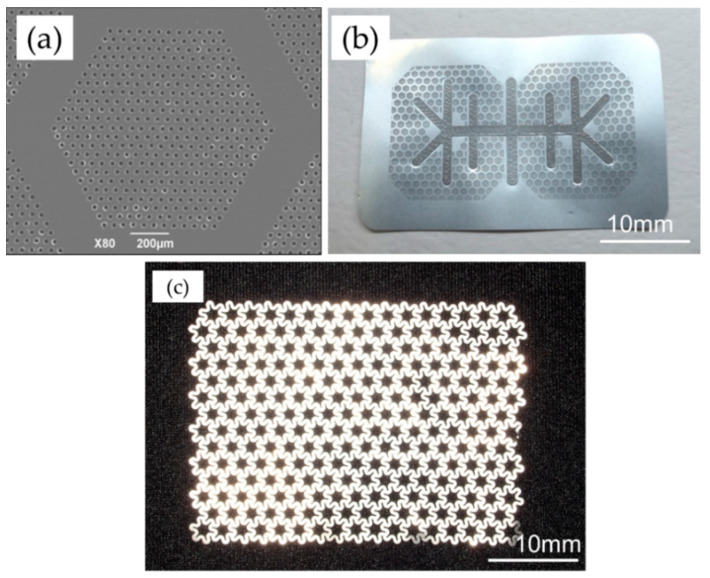
Membrane. (**a**) Ti-honeycomb membrane (TiHM). Pores with diameters of 20 μm were created in a pure titanium sheet with a 20-μm thickness, a 50-μm pitch, within regular hexagonal compartments that had an inscribed 1.0-mm-diameter circle. These processed sections were arranged together with unprocessed, 0.2-mm-wide sections in a honeycomb shape. (**b**) TiHM (L size, 27 mm × 42 mm with a frame). A titanium reinforcement frame is jointed on the membrane. (**c**) Ultra flex mesh plate (UFMP, 35 mm × 48 mm). Star-like pores were created, and featured an inscribed, ca., 1600-μm-diameter, circle.

**Figure 3 materials-14-03358-f003:**
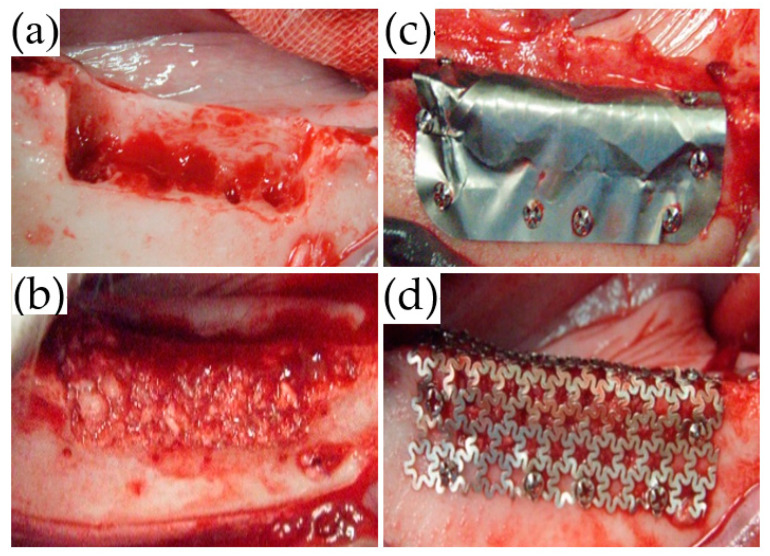
Surgical procedure. (**a**) A box-shaped defect was created with a 25-mm mesiodistal width, 5-mm buccolingual width, and 6-mm depth. (**b**) The harvested iliac bone was pulverized and grafted to the bone defect. (**c**) The bone defect was covered with TiHM and secured with eight titanium screws. (**d**) The bone defect was covered with UFMP, eight titanium screws were used to secure it, as performed with TiHM.

**Figure 4 materials-14-03358-f004:**
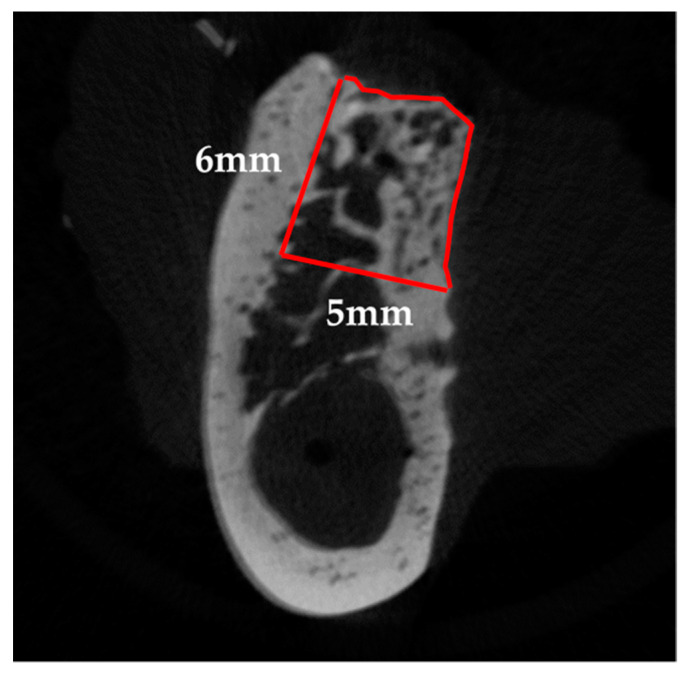
Radiographic analysis. The red line represents the region of interest (ROI). On the sections, 12 mm away from the mid-section, both mesially and distally, the range surrounded by the linear margin of the bone defect (6-mm height, 5-mm base) and the perimeter of the newly formed bone was manually outlined after identifying the border between the bone defect and the host bone.

**Figure 5 materials-14-03358-f005:**
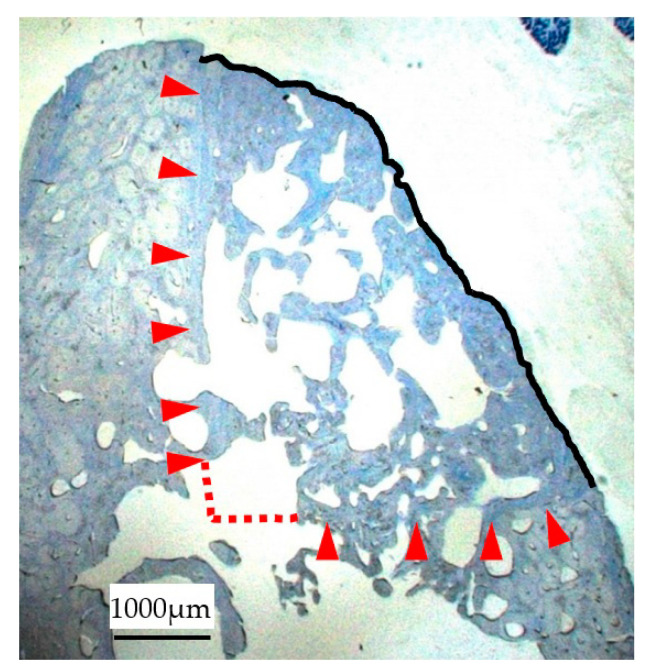
RBA measurement. The acquired image of the TB-stained sections was used to measure RBA, which is done by tracing the perimeter of the newly formed bone (black line) as well as the border between the original host and regenerated cortical bone, as determined by the shading difference in the TB staining (red arrows).

**Figure 6 materials-14-03358-f006:**
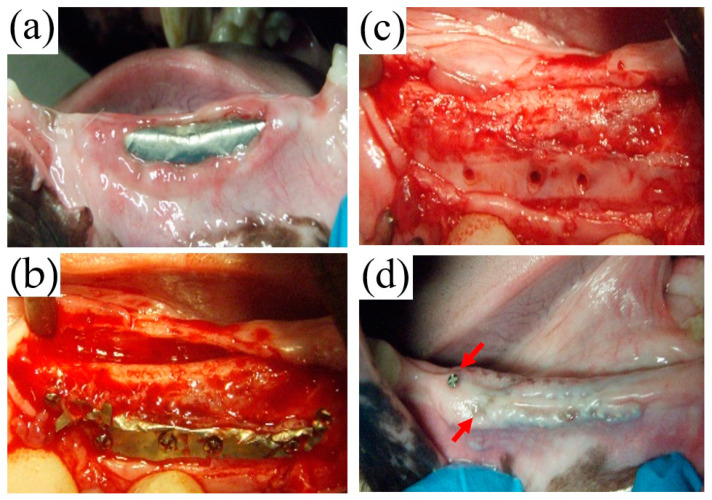
Macroscopic observation of the membrane exposure. (**a**) The membrane exposure in the TiHM group. The membrane exposure was found on the alveolar crest at 3 weeks postoperatively. (**b**) In the same site as (**a**), the membrane was extracted at 9 weeks postoperatively. (**c**) In the same site as (**b**), after removing the granulation tissue, the new bone was found beneath the tissue. (**d**) The membrane exposure in the UFMP group. The secured screw and the surrounding membrane exposure were found on the alveolar crest at 8 weeks postoperatively (red arrows). However, the removal was not required because the exposures were not changed.

**Figure 7 materials-14-03358-f007:**
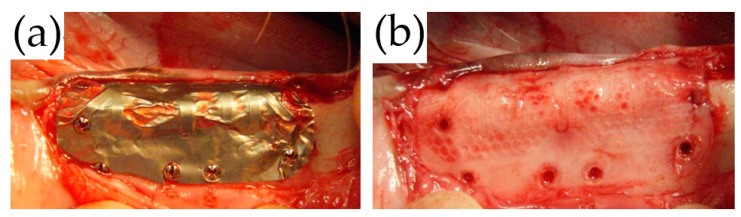
Macroscopic observation at membrane removal in the TiHM group. (**a**,**b**) There was no adhesion of the membrane to the surrounding tissue, and the membrane was able to be peeled back with ease. The bone surfaces in contact with the laser-processed sections showed unprocessed sections and honeycomb-shaped compartments, which were clearly colored red.

**Figure 8 materials-14-03358-f008:**
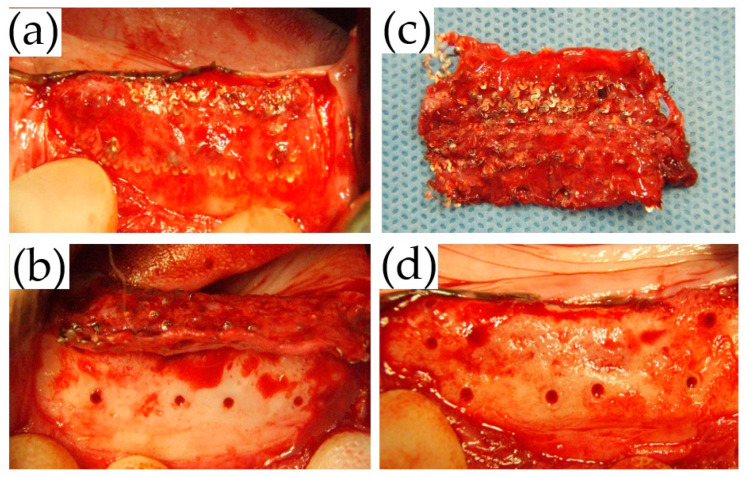
Macroscopic observation at membrane removal in the UFMP group. (**a**,**b**) It was difficult to elevate the mucoperiosteal flap, as well as the tissue beneath the membrane from the membrane because of the adhesion. (**c**) The membrane was extracted with the surrounding scar tissue. (**d**) The surface was replaced with the bone tissue.

**Figure 9 materials-14-03358-f009:**
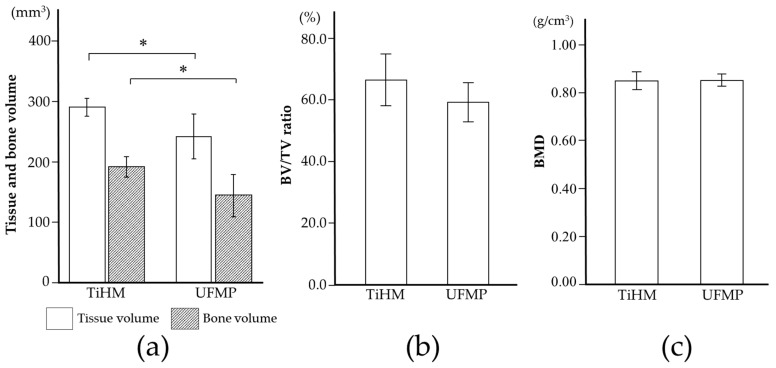
Radiographic analyses (**a**) TV and BV. (**b**) BV/TV ratio. (**c**) BMD. All data are presented as means ± SD; * *p* < 0.05.

**Figure 10 materials-14-03358-f010:**
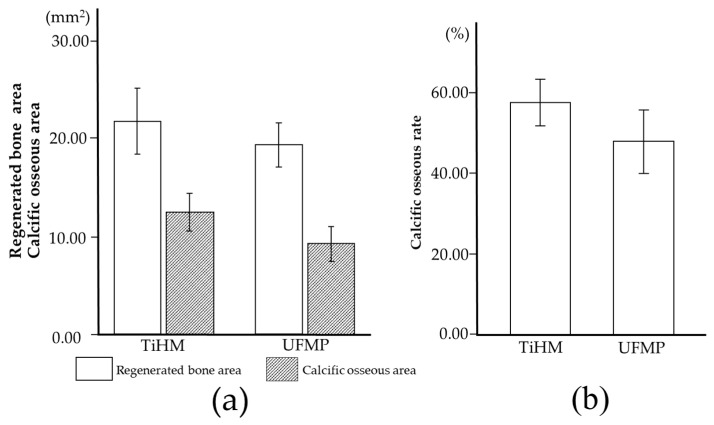
Histomorphometric analyses (**a**) RBA and COA. (**b**) COR. All data are presented as means ± SD.

## Data Availability

The data presented in this study are available on request from the corresponding author.

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
