# Peer review of "Comparing the Efficacy of a Microperforated Titanium Membrane for Guided Bone Regeneration with an Existing Mesh Retainer in Dog Mandibles"

_materials, 2021, doi:10.3390/ma14123358_

Round 1

Reviewer 1 Report

The paper is interesting and even if the mesh technique in less in use than in the past due to the exposure complication, the new titanium barrier seems useful.

The manuscript “Comparing the Efficacy of a Microperforated Titanium Membrane for Guided Bone Regeneration with an Existing Mesh Retainer in Dog Mandibles” presents a study comparing the use of two titanium membranes for bone regeneration in acute-type lateral ridge defects in dogs.

In the manuscript, the results are interpreted appropriately; all conclusions are justified and supported by the results; the article is written in an appropriate way; the study is correctly designed and technically sound; the analyses are performed with the highest technical standards; the methods are described with sufficient details to allow another researcher to reproduce the results; the conclusions are interesting for the readership of the Journal; there is an overall benefit to publishing this work.

The paper is interesting and even if the mesh technique is less in use than in the past due to the exposure complication, the new titanium barrier seems useful.

Line 54: why "on the other hand"?

The paper needs a language review by a mother-language.

The plagiarism analysis detected a 15% of text similar to www.quintpub.com/.../omi/omi_34_5_Hasegawa_p1132.pdf

Maybe it's better to change the sentences. I attach the report.

Statistical analysis: please specify how sample size was calculated.

Please specify in the results section standard deviation among with means.

I think that the paper, after review, can be considered for publication

For the reasons listed above, my final recommendation is to accept after minor revisions the manuscript.

Author Response

Response to Reviewer 1 Comments

  • Point 1: Line 54: why "on the other hand"?

The paper needs a language review by a mother-language.The plagiarism analysis detected a 15% of text similar to www.quintpub.com/.../omi/omi_34_5_Hasegawa_p1132.pdf. Maybe it's better to change the sentences. I attach the report.

Response 1:

Thank you for your suggestion. We now have had professional native English-speaking editors proofread our revised manuscript.

Although we could not find your attached report, we assumed that the paper that was detected your plagiarism analysis was our previous published paper, “Hasegawa H, et al., Evaluation of a newly designed microperforated titanium membrane with beta-tricalcium phosphate for guided bone regeneration in dog mandibles. Int J Oral Maxilofac Implants 34, 1132-1142, 2019.” Although this previous study used similar materials and methods, including the dog mandibular model that was used in the present study, these two studies are completely different. However, a similar sentence has been found in this manuscript, which we changed in line 54-58.

 FRIOS® Boneshield (FBS; Dentsply Sirona Corp., Tokyo, Japan), which is a thin film-like titanium membrane, has been used globally as a more flexible retainer for enveloping the filling. With a thickness of approximately 20 μm and micropores (diameter, 40–60 μm; pitch 250 μm) that allow the free diffusion of nutrients and tissue fluid, FBS is easy to handle because of its shape and is significantly effective in the clinical setting [9–13].

  • Point 2: Statistical analysis: please specify how sample size was calculated.

Response 2:

Before this experiment started, we calculated the needed sample size using power analysis (G*power3.1, free software, effect size, 1.0; αerr probability, 0.05; power, 0.8). The required sample size for a t-test was greater than 17 individuals. This large sample study had the ethical and cost problems. Therefore, the present study was performed as a pilot study with small sample. As the result, significant differences were found in regenerated tissue and bone volume. Furthermore, in these two indices, the non-parametric Mann-Whitney test, also showed significant differences (asymptotic significance probability= 0.043, each in both indices). Based on the above observations, we chose not to specify how the sample size was calculated. However, we have added “(N=4 each)” in the 2.11. statistical analysis section (line256) to specify the sample size.

We would like to perform the further study with required sample size to define the optimal features including pore size in a titanium membrane for a GBR procedure.

Point 3: Please specify in the results section standard deviation among with means.

Response 3:

Thank you for your suggestion. We have added the standard deviation in the results, line 300-313:

“3.2. Radiographic Evaluation

The mean TV, BV, BV/TV ratio, and BMD in the TiHM group were 290.4 ± 14.3 mm3, 192.1 ± 16.3 mm3, 66.45 ± 8.4%, and 0.85 ± 0.04 g/cm3, whereas in the UFMP group were 242.0 ± 37.0 mm3, 144.4 ± 35.0 mm3, 59.2 ± 6.4%, and 0.85 ± 0.03 g/cm3, respectively. Thus, the TV, BV, and BV/TV ratio were larger in the TiHM group than in the UFMP group (Figure 9a, b). Furthermore, the former two indices showed significant differences (p = 0.05; p = 0.049). However, the BMD value was similar in both the groups (Figure 9c).”

“3.3. Histomorphometric Analysis

As shown in Figure 10, the respective mean RBA, COA, and COR were 21.8 ± 3.4 mm2, 12.5 ± 1.9 mm2, and 57.4 ± 5.8% in the TiHM group and 19.4 ± 2.3 mm2, 9.3 ± 1.8 mm2, and 47.9 ± 7.9% in the UFMP group, respectively. The values were larger in the TiHM group than in the UFMP group, although no statistical differences were found.”

Reviewer 2 Report

The study comparing the efficacy of a microperforated titanium membrane with an existing mesh retainer for guided bone regeneration in animals might be interesting especially considering the possible future validation and clinical applications in humans.My concerns and suggestions are listed below.

Introduction section:

Please, remove the word "and" (line 42).

I would suggest tosubstitute "using" (line 75) with "in".

Materials and Methods:

"Fabrication of microperforated titanium membrane paragraph" should be re-written in order to be more clear (line 78).

Did the Authors perform the fabricate the membrane? If so, please specify it in the text.

Please, explain how the box-shaped defects were standardized (line 127).

Discussion:

Please, explain or express better what you meant with "perform good cleaning" (line 355).

Please, expand on “Therefore, even though membrane exposure, particularly early exposure, can inhibit bone regeneration, it does not necessarily lead to complete bone loss” (lines 402-403) and add the related references.

Author Response

Response to Reviewer 2 Comments

Point 1: Please, remove the word "and" (line 42).

Response 1:

Thank you for your suggestion. We remove the word “and” in line 42.

“The chief requirements for the membrane are described as the following: (1) biocompatibility; (2) mechanical properties for retaining the filling and a spacemaking to create a space to osteogenic regeneration; (3) clinical manageability; (4) cell occlusivity; (5) tissue integration for stability with respect to the surrounding tissue [2,3].”

Point 2: 2, I would suggest tosubstitute "using" (line 75) with "in".

Response 2:

Thank you for your suggestion. We corrected the sentence in line 75.

“Therefore, it is necessary to investigate the efficacy of these two membranes with different thicknesses and pore sizes. In the current study, the efficacies of TiHM and titanium mesh were evaluated in alveolar bone defects in the mandible of beagle dogs accompanied with autologous bone grafts.”

Point 3: "Fabrication of microperforated titanium membrane paragraph" should be re-written in order to be more clear (line 78). Did the Authors perform the fabricate the membrane? If so, please specify it in the text.

Response 3:

Thank you for your suggestion. We re-wrote the sentence in line 79-113. Actually, we designed the dense and sectioned microperforated TiHM structure. The sample fabrication was performed by Lastech Co., Ltd. and Hayashi Seiki Seizou Co., Ltd.

“2.1. Fabrication of a microperforated titanium membrane

A microperforated TiHM structure (size L, 25 mm × 35 mm with a frame) was designed to be formed into a precise and unified piercing array on a thin titanium sheet. The ablation piercing process used a short pulsed laser with a nanosecond pulse duration, to produce a hexagonal pored section that includes a uniform, staggered array of ordered mircon-sized holes (Figure 1). The irradiation power and focus of the laser beam were optimized to penetrate through the thin pure titanium sheet as quickly as possible, to accelerate serial perforation and prevent thermal degeneration of the material (Lastech Co., Ltd., Fujimi, Japan). A heat-affected zone along the cut edge of the round hole, which was produced after the laser process, remained as a thin layer of dross. Thus, after completing the piercing process, the workpiece was etched by immersion into a hydrofluoric acid solution.

 A conventional laser processing system that used an oscillating, nanosecond pulse frequency light source was optically tuned to confirm an output with a focused diameter that was within 20 microns. Galvano-drive mirrors reflected and emitted the beam to the workpiece and oriented the beam to scan the surface area over a range of a few square millimeters. The penetration processing velocity with an ø 20 µm precise diameter laser was 80 holes per second. When the Galvano-drive mirror serial piercing process completed over the scannable range, a motorized X-Y stage moved the workpiece horizontally by a few millimeters. Then, laser piercing process was performed under programmable control in the neighboring area.

The sample material was manufactured with dense microperforations (diameter, 20 μm; pitch, 50 μm) within regular hexagonal non-perforated compartments (inscribed diameter, 1.0 mm; width, 0.2 mm), on a pure titanium 20 μm-thick sheet with a honeycomb shape (Figure 2a, b). The titanium reinforcement frame that imparted maintenance properties for the spacemaking was joined on the membrane by multiple spot welding (Hayashi Seiki Seizou Co., Ltd., Sukagawa, Japan).

An ultra-flex mesh plate (UFMP, 35 × 48 mm, 100-μm thickness, Kyocera Co., Kyoto, Japan) was used as the standard thinner titanium mesh in this study and had star-like pores with a circumscribed diameter of approximately 3200 μm, inscribed diameter of approximately 1600 μm, and thickness of 100 μm. These pores enabled rigid screw fixation (Figure 2c).”

Point 4: Please, explain how the box-shaped defects were standardized (line 127).

Response 4:

We made the box-shaped defects in the same size precisely after extracting tooth. The method was described in line144-152:

“Mucosal incisions were made: one along the cervical region of P1 to M2, followed by longitudinal incisions at P1 and M2. The mucoperiosteal flap was elevated, and the second premolar (P2), third premolar (P3), fourth premolar (P4), and first molar (M1) were extracted. Subsequently, using a dental fissure bur (Osada portable unit “Daisy”, Osada Electric Industry Co., Ltd., Tokyo, Japan), a box-shaped defect (mesiodistal width, 25 mm; depth, 6 mm; depth from buccoalveolar bone surface, 5 mm) was created while continuing irrigation with physiological saline (Figure 3a).”

Point 5: Please, explain or express better what you meant with "perform good cleaning" (line 355).

Response 5:

 Thank you for your suggestion. We have amended the sentence in line 356, “This exposure was probably due to technical failures of suturing and may have enlarged because the animal was unable to rest adequately and perform frequent mouth care and cleaning.”

Point 6: Please, expand on “Therefore, even though membrane exposure, particularly early exposure, can inhibit bone regeneration, it does not necessarily lead to complete bone loss” (lines 402-403) and add the related references.

Response 6:

Thank you for your suggestion. We have amended the sentence and added the references in line 405, “Therefore, even though membrane exposure, particularly early exposure, can inhibit bone regeneration, it does not necessarily lead to complete bone loss, and the membrane does not necessarily require removal, as described above [5, 6, 9, 10, 21–24].

Reviewer 3 Report

Journal: Materials

Manuscript ID: materials-1243309

Title: Comparing the Efficacy of a Microperforated Titanium Membrane for Guided Bone Regeneration with an Existing Mesh Retainer in Dog Mandibles

Authors: Hiroshi Hasegawa *, Tetsuharu Kaneko, Manabu Endo, Chihiro Kanno, Morio Yamazaki, Sadanoshin Yaginuma, Hiroki Igarashi, Hideaki Honma, Seiichiro Masui, Mizuki Suto, Yukihiko Sakisaka, Hiroshi Ishihata

Submitted to section: Biomaterials

General Comments:

In the present study, a microperforated titanium membrane (Ti Honeycomb Membrane, TiHM) was evaluated for its efficiency compared with a conventional standard titanium mesh in the reconstruction of mandibular bone defects with autologous bone graft in dogs. Results showed that the regenerated bone volume was significantly greater in the TiHM group than in the control group. Although the area of regenerated bone and the area of calcified bone were larger in the TiHM group than in the control group, the differences were not significant. Bone mineral densities were comparable in both groups.

This is an interesting study comparing the effectiveness of both membranes with different thicknesses and pore sizes. The experiments are well performed and results are well demonstrated. In clinical applications, it has been proposed that mesh can be used more for large bone augmentations, while TiHM is suitable for small to medium bone augmentations. The work is a worthy contribution to the field.

Minor point:

Line 80: "structuire" should be corrected as "structure"

Author Response

Response to Reviewer 3 Comments

Point 1: Line 80: "structuire" should be corrected as "structure".

Response 1:

Thank you for your suggestion. We have corrected it in line 79, “A microperforated TiHM